# Tensiomyographic Neuromuscular Response of the Peroneus Longus and Tibialis Anterior with Chronic Ankle Instability

**DOI:** 10.3390/healthcare9060707

**Published:** 2021-06-10

**Authors:** Tsubasa Tashiro, Noriaki Maeda, Junpei Sasadai, Somu Kotoshiba, Shogo Sakai, Yuta Suzuki, Hironori Fujishita, Yukio Urabe

**Affiliations:** 1Department of Sports Rehabilitation, Graduate School of Biomedical and Health Sciences, Hiroshima University, Hiroshima 734-8553, Japan; tsubasatashiro716@hiroshima-u.ac.jp (T.T.); norimmi@hiroshima-u.ac.jp (N.M.); ksomu0202@hiroshima-u.ac.jp (S.K.); shg.sakai@gmail.com (S.S.); 2Sports Medical Center, Japan Institute of Sports Sciences, Tokyo 115-0056, Japan; jumpei.sasadai@jpnsport.go.jp; 3Department of Rehabilitation, Matterhorn Rehabilitation Hospital, Hiroshima 737-0046, Japan; yt.suzuki28@gmail.com; 4Sports Medical Center, Hiroshima University Hospital, Hiroshima University, Hiroshima 734-8551, Japan; h-fujishita@hiroshima-u.ac.jp

**Keywords:** tensiomyography, chronic ankle instability, muscle contractile response, peroneus longus, tibialis anterior

## Abstract

This study aimed to investigate the muscle contractile response of the peroneus longus (PL) and tibialis anterior (TA) in groups with and without chronic ankle instability (CAI) using tensiomyography. Twenty-three adults, 12 with CAI and 11 healthy participants, participated in this study. All subjects underwent a tensiomyographic assessment of the PL and TA to measure delay time, contraction time and maximal displacement. The ankle evertor and invertor normalized peak torques, maximum work done and muscle thickness of the PL and TA were calculated. The delay time and contraction time of the PL in the CAI side were significantly higher than those in the healthy group (*p* < 0.05); however, no significant difference could be detected in the TA between groups. Furthermore, there was no significant difference in the normalized peak torques, maximum work done and muscle thickness of the PL and TA between groups. The CAI side demonstrated a delayed muscle contractile response of the PL when compared with the healthy group although there was no difference in muscle strength and muscle size. Clinicians should consider the muscle contractile response of the PL for rehabilitation of the ankle evertor with CAI.

## 1. Introduction

Lateral ankle sprains are one of the most frequent injuries in trauma [1,2]. Many individuals may experience symptoms such as pain, swelling, impairment of ankle function, a sensation of “giving way” in the ankle and repeated sprains for a long duration after the first ankle sprain [3,4,5]. Acute ankle sprains occur very frequently and chronic ankle instability (CAI) is secondary to them [2,6]. According to previous studies, a high proportion of individuals with ankle sprains would eventually have CAI [7]. Lateral ankle sprains occur due to excessive inversion loading of lateral tissues of the ankle [8] and are frequently caused by the enforced inversion of the ankle, regardless of the plantar or dorsal flexion of the ankle [9,10]. The peroneus longus (PL) is one of the muscles that reacts and contracts to rapid inversion stresses of the ankle and is therefore essential in providing dynamic stability control of the lateral ankle joint tissues. Although many investigators have focused on the ankle evertor strength with a gold standard isokinetic procedure and the muscle thickness of the PL using ultrasound, no difference was reported in these parameters between individuals with and without CAI [5,11]. On the other hand, disturbances in the neurotransmitter system including the delayed muscle response of the PL are known to be a marked symptom in cases of CAI and have been suggested as a possible mechanism of functional instability after lateral ankle sprains [12].

Unintentional ankle perturbations such as a trapdoor with the use of electromyography (EMG) is a popular way to test the function of muscle responses in individuals with CAI [13,14]. Such experimentation includes the PL and tibialis anterior (TA) muscles, which can be damaged from overly supinating the ankle. In a systematic review, the muscle reaction time of the peroneal muscles in patients with CAI was significantly delayed compared with that in healthy subjects [15]. Researchers have considered that a delayed muscle reaction time in the peroneal muscle may be a risk factor for recurrent ankle sprains. In addition, previous studies have reported that the distal muscles (PL and TA) showed a delayed onset of muscle activity than the proximal muscles (gluteus maximus, gluteus medius) in individuals with CAI, resulting in altered muscle activity patterns [16]. Although EMG measures electrical activity during voluntary muscle action, it does not measure the mechanical muscle contractile response of a particular muscle. As this method involves detecting neuromuscular responses to a large number of somatosensory feedbacks, it can be difficult to identify the physiological mechanism that resulted in the response.

Tensiomyography (TMG) has recently been applied to evaluate the mechanical characteristics of skeletal muscles in a convenient and non-invasive way [17]. With the use of this method, it is possible to evaluate several contraction parameters including contraction time (Tc), delay time (Td) and maximum radial displacement (Dm). Tc (calculated from the time from 10% to 90% of Dm) refers to the speed of the generating force, Td denotes the speed of the muscle conduction and Dm represents the stiffness of the muscle belly [18]. TMG enables an assessment of involuntary muscle contractile responses and muscle belly displacement after electrical stimulation to individual muscles.

However, to the best of our knowledge, there are no studies that have examined the muscle contractile response of the lower extremities of CAI subjects with TMG. The purpose of the present study was to investigate the muscle contractile response in the PL and TA muscles of subjects with CAI. Our hypothesis was that the PL of the CAI group would be lacking muscle response and contraction compared with the healthy group.

## 2. Materials and Methods

### 2.1. Study Design and Setting

We conducted a cross-sectional study to investigate neuromuscular responses on the CAI and non-CAI side of subjects with CAI. This study was done in the laboratory of Hiroshima University. The participants in this study included physically active university students. All dependent variables including the TMG scores for the PL and TA, the evertor and invertor mean normalized peak torques and the mean normalized maximum work done at 60°/s and 180°/s and the muscle thickness for the PL and TA were compared between the CAI and non-CAI sides of the CAI group and the healthy group (the independent variables).

### 2.2. Subjects

Twelve young subjects (11 men and 1 woman) with CAI and 11 healthy subjects (9 men and 2 women) with no history of a lateral ankle sprain were included in this study (Table 1). In the CAI group, all participants fulfilled the following inclusion criteria as recommended by the International Ankle Consortium [19]: (1) the first ankle sprain occurred over a year ago, (2) no history of a sprain within 6 weeks before testing and (3) a Cumberland Ankle Instability Tool (CAIT) score of less than 24. The participants in the healthy group were required to have a CAIT score of 27 or higher. The subjects were excluded from the study if one of the following was reported: (1) previous surgery on a lower extremity or (2) previous disease or pathology that could affect neuromuscular control or interfere with the ability to perform a complete examination [19]. A personal report of the number of lateral ankle sprains was collected as other descriptive data. No statistically significant differences in age, height or mass were found between the groups. All variables were tested on the CAI and non-CAI sides of the CAI group and on both legs of the healthy group.

This study was authorized by the Institutional Review Board at Hiroshima University (ID number: E-1905) and all of the subjects provided signed informed consent to participate in this study.

### 2.3. Procedures

#### 2.3.1. Tensiomyography (TMG) Assessment

The method of measuring TMG was the same for both groups. The investigator, T.T., with three years or more of TMG experience, took all of the measurements. A portable device was percutaneously applied to generate electrical stimulation, which induced a muscle contraction that was detected by a digital transducer placed on the muscle belly. The delay time (Td), contraction time (Tc) and muscle displacement (Dm) of the PL and TA were collected by TMG (TMG Measurement System; TMG-BMC Ltd., Ljubljana, Slovenia) [20]. The TMG variables were obtained from the CAI and non-CAI sides in the CAI group and from both legs in the healthy group. The PL was measured with the participants in the prone position and the TA was measured with the participants in the supine position. A foamed cushion was utilized in the supine position to immobilize the knee joint flexion angle at 30° [21]. The radial muscle displacement was measured vertically across the muscle belly using a digital transducer Dc–Dc Trans-Tek^®^ (GK 40, Panoptik d.o.o., Ljubliana, Slovenia). Based on a study by Delagi et al., the anatomical position of the sensor was normalized to all subjects [22]. The self-adhesive electrodes (TMG electrodes, TMG-BMC d.o.o. Ljubljana, Slovenia) were located at equal distances from the measurement point, proximal (anode) and distal (cathode) of the sensor. Electrical stimulation was performed using a TMG-100 System electro stimulator (TMG-BMC d.o.o., Ljubljana, Slovenia) at a pulse duration of 1 ms and an initial amplitude of 50 mA. For all trials, the amplitude was increased gradually by 10 mA until no further increase in Dm was observed at the peak stimulus output (110 mA). The PL and TA were examined twice: the first trial assured the proper working for TMG and a definitive value was obtained in the second trial. In previous research, the intraclass correlation coefficient (ICC) test-retest values of Td, Tc and Dm ranged from 0.91 to 0.98, corroborating the outstanding reliability of TMG [23]. Our results showed that the ICC (1, 3) of Td, Tc and Dm were 0.98, 0.99 and 0.99 in the PL, respectively, and 0.96, 0.99 and 0.97 in the TA, respectively.

#### 2.3.2. Muscle Strength Measurement

We investigated the isokinetic evertor and invertor mean normalized peak torque (Nm) using the Biodex System 4 (Biodex Medical Systems Inc., Shirley, NY, USA). Each subject sat on the adjustable seat of the dynamometer and elevated the test leg with the support arm below the knee. The foot was positioned on a footboard and fixed by two Velcro straps. All of the subjects underwent the test barefoot. The subtalar joint was placed in a neutral position by aligning the axis of the dynamometer to cross the sagittal axis of the joint. A rubber heel cup provided stability on the footboard and adjustable thermoplastic stays were put on the medial and lateral boundaries of the foot. A pair of standard Velcro straps placed diagonally held the trunk steady and a single strap fixed the hip. The arms were placed crossed on the chest and the opposite leg was put on the support arm attached to the chair. The ankle joint range of motion was measured with the subject on the dynamometer. The setting of the end range was normalized for all test subjects from a 40° eversion to a 45° inversion. To eliminate the influence of gravitational effects on the data, the weight of the leg of each subject was measured and the data were corrected within the Biodex software. Isokinetic testing of the evertor and invertor mean normalized peak torque was performed at a velocity of 60°/s and 180°/s with a concentric contraction of the ankles. A 2 min interval was provided between the warm-up and the test. The subjects were instructed to test their best effort. All subjects were tested five times and the evertor and invertor mean normalized peak torques and mean normalized maximum work done were recorded by the Biodex software. Each value calibrated with body weight (Nm/kg) was employed during the data analysis to correct for body weight.

#### 2.3.3. Ultrasound Measurement

An ultrasound unit (Noblus; Hitachi Aloka Medical Japan, Tokyo, Japan) with an 8 to 12 MHz range linear transducer, 8 MHz frequency, 4 cm image depth and 1.5 cm depth of focus was used to longitudinally visualize and obtain images of the muscle thickness in the PL and TA. Each subject was evaluated in the prone position for the PL muscle and in the supine position for the TA muscle. The PL muscle thickness images were captured at 50% of the distance from the fibular head to the lateral malleolus [24,25]. The thickness of the TA was measured at 20% of the distance from the fibular head to the inferior border of the lateral malleolus [24]. During the measurement, proper contact of the probe with the skin was ensured. Three images were finally obtained for each muscle. All stored ultrasound images were recorded by an investigator, T.T., who has more than three years of ultrasound experience. The muscle thicknesses of the PL and TA from each image were analyzed using Image J software (National Institute for Health, Bethesda, MD, USA) by an investigator, S.K., who was blinded to group assignments. For both the PL and TA, the ICC (1, 3) of the muscle thickness was 0.99.

### 2.4. Statistical Analyses

All outcomes were analyzed using a normal analysis with a Shapiro–Wilk test. An unpaired t-test was used to compare the demographic information of the subjects between the CAI and healthy groups. The comparisons of the TMG value, mean normalized peak torque, mean normalized maximal work done, muscle thickness of the PL and TA between the CAI side and non-CAI side and healthy subjects were conducted using a one-factor ANOVA test. The effect sizes were calculated using eta-squared (η^2^) statistics and the post-hoc observed power was generated by G*Power 3.1 (Kiel University, Kiel, Germany). For the follow-up, a Tukey post-hoc analysis was carried out to statistically analyze the difference between the CAI side, non-CAI side and the healthy group. Cohen’s d statistics were calculated as the effect sizes for post-hoc comparisons. The significance level was defined at 5% (*p* < 0.05). SPSS version 23.0 for Windows (SPSS Inc., Chicago, IL, USA) was used for analyzing the data. A pre-study sample size was estimated from the Td of the PL of three legs in the CAI side (mean 18.66), three legs in the non-CAI side (mean 16.30) and three legs in the healthy group (mean 16.12) with an alpha level *p* < 0.05 and power of 0.8 using G*Power 3.1 because there was no previous study that used the TMG value of the PL as an outcome. A minimum of 33 subjects was calculated to be needed.

## 3. Results

The study ultimately included 12 subjects with CAI and 11 healthy subjects. Of the demographic characteristics, there were significant differences only in the CAIT and number of lateral ankle sprains in both groups. The comparison of the TMG values of the PL and TA between the CAI and healthy groups are shown in Figure 1. A significant difference was found between the groups in the PL and a post-hoc analysis showed that the CAI side had a significantly higher Td and Tc than the healthy group. However, there was no significant difference in the Dm of the PL and Td, Tc and the Dm of the TA.

The comparisons of evertor and invertor mean normalized peak torques and the mean normalized maximum work done at 60°/s and 180°/s between the CAI and healthy groups are presented in Table 2. No statistically significant difference in the mean normalized peak torques and the mean normalized maximum work done was identified between the two groups. The comparison of the muscle thickness between the CAI and healthy groups is shown in Table 3. The results were statistically insignificant for the muscles in both groups.

## 4. Discussion

The most important result of this study was that the Td and Tc of the CAI side were significantly higher than those of the healthy group in the PL, even though there were no differences in the muscle strength or muscle size. In contrast, no significant difference in any of the TMG values of the TA could be found among the two groups. According to our knowledge, this is the first study to compare the TMG value of the PL and TA between CAI and healthy populations. This novel investigation of the muscle contractile response in CAI subjects can provide insights for therapists and researchers who treat patients with lateral ankle sprains and CAI.

In the present study, the TMG values of the PL in the CAI side were significantly higher in Td and Tc compared with those of healthy subjects. Previous investigations have shown that Td indicates the muscle fiber conduction velocity. Tc is the time from the onset of the muscle contraction to the peak contraction and it reflects the generation of the speed of force [26,27]. Considering these reports, the PL in the CAI side possibly had a lesser ability to generate the muscle force quickly on the muscle contraction (increase in Td or Tc) compared with healthy subjects. The PL is the main operating muscle of ankle eversion; it has a stabilizing mechanism of the lateral part of the ankle joint and prevents lateral ankle sprains by contracting defensively against a sudden ankle inversion. There are several reports on delayed reaction times in the PL of patients with CAI measured using EMG. Although peroneal tendon injuries often occur in conjunction with lateral ankle sprains [12], stretched muscle trauma injury can affect the contractility and flexibility of the muscle-tendon component and impair the ability to generate muscle force quickly [28]. The characteristics of this lateral ankle sprain injury may be associated with increased Td and Tc in the CAI side. In addition, there was no statistically significant difference in TMG values between the CAI and non-CAI side. This result was comparable with the EMG study, which found no difference in the neuromuscular responses of the PL and TA between stable and unstable ankles [29]. The fact that mechanical ankle instability was not evaluated might be a factor in this result. Additionally, there was no significant difference in the TMG values of the TA muscle between the CAI side, non-CAI side and the healthy legs. This result suggested that the muscle contractile response in the PL might be impaired after lateral ankle sprains. Irrespective of the cause, clinically important changes in the muscle contractile response of the PL muscle in the CAI group might imply that rehabilitation programs should include functional exercises to help overcome this deficit. Thus, prospective studies are required to determine the extent to which an altered muscle contractile response influences future injury risk for CAI.

There were no significant differences between the mean normalized peak torques and the mean normalized maximum work done of the ankle evertor and invertor at both 60 and 180 degrees between the CAI and healthy groups. As a lateral ankle sprain is due to an over-inversion of the ankle, most investigators involved in the study of CAI have given attention to the ankle evertor strength [30]. Cho et al. showed that the isokinetic strength of the ankle evertor in the CAI group was significantly weaker than that in the control group [31]. Similarly, Donnelly et al. observed less ankle evertor strength in the CAI group compared with the healthy group [32]. However, it was shown in several reports that there was no significant difference in the ankle evertor strength between CAI and healthy subjects [4,5,30]. The evertor strength is created by the joint action of several muscles. If the PL does not function properly owing to a dysfunction caused by an ankle sprain, other muscles may compensate for it. Ahn et al. reported that CAI subjects showed compensatory activity of the extensor digitorum longus upon ankle eversion and, in contrast, decreased muscle activity of the PL [33]. This could be explained by the fact that muscle strength does not weaken even if there are defects in the muscle contractile response. Our results were in line with recent studies using TMG such as the study conducted by Maeda et al. [34]. They reported that the muscle contractile response was impaired even when the isokinetic peak torque was not reduced in patients after reconstruction surgery of the anterior cruciate ligament in chronic cases. Taken together, assessments of the muscle contractile response as well as muscle strength and muscle activity are required. Moreover, we must consider whether an impaired muscle contractile response could be a factor for CAI.

No differences between the CAI and healthy groups were observed for the muscle thickness of the PL and TA. Our results were similar to those of studies that evaluated muscle volume and strength of the lower legs through magnetic resonance imaging in subjects with/without CAI [35]. Additionally, Koldenhoven et al. reported no significant difference in the cross-sectional area of the PL in patients with or without CAI [25]. The investigators suggested that this result may be due to changes in the neuromuscular function as opposed to muscle volume [25,35] and that may account for the absence of differences in the PL and TA. Furthermore, a previous intervention study reported that a peak amplitude was associated with muscle thickness [36]. In the present study, Dm and muscle thickness were not significantly different between the CAI and healthy groups. If there was no difference in the muscle thickness between the groups, the peak amplitude during the muscle contraction might not be very different. Therefore, the lack of difference in the muscle thickness between the PL and TA was consistent with the TMG results (Dm). Our current results suggested the need to focus on both the muscle volume and neuromuscular function during the evaluation of CAI.

This study has a few limitations. The first limitation was the small sample size. Experiments with a sufficient number of subjects are necessary for future research. Second, all measurements were performed in stationary conditions and therefore differed from dynamic characteristics on the field. Third, we focused on the PL as an evertor muscle and the TA as an invertor muscle but other muscles also perform these movements. Fourth, it is important to note that this was a cross-sectional study; thus, it is unclear whether the larger Td and Tc in the PL were the result of CAI or whether athletes with a larger Td and Tc in the PL are predisposed to CAI. Prospective studies may provide an answer to this question. Given that current research has identified muscle contractile response deficits in CAI individuals primarily through methods that confirm the muscle contractile response to the direct electrical stimulation of the skeletal muscle, advanced experiments investigating particular neuromuscular functions in subjects with CAI could be insightful.

## 5. Conclusions

In conclusion, the present findings demonstrated the impaired muscle response and muscle contraction of the PL, which indicated the requirement for ankle evertor training for CAI. Furthermore, the clinical implications of this study suggested the significance of rehabilitation of the muscle response and contraction speed. Additional studies are required to determine the appropriate rehabilitation program for improving the muscle contractile response in CAI.

## Figures and Tables

**Figure 1 healthcare-09-00707-f001:**
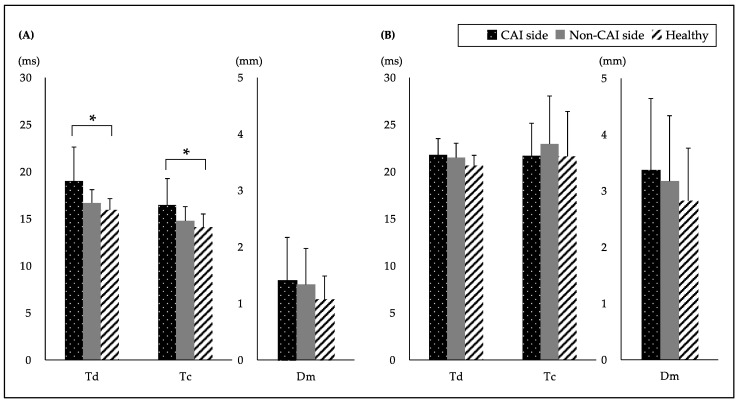
Comparison of the TMG values between the CAI and healthy groups of subjects. TMG values of the (**A**) peroneus longus (PL) and (**B**) tibialis anterior (TA) are shown. Data are expressed as the mean ± SD. * *p* < 0.05. (ms) means the time. (mm) means the displacement. CAI: chronic ankle instability; Td: delay time; Tc: Contraction time; Dm: maximal displacement. One-factor ANOVA test: Td of the PL, *p* = 0.033, η^2^ = 0.25, observed power (OP) = 0.92; Tc of the PL, *p* = 0.025, η^2^ = 0.21, OP = 0.70; Dm of the PL, *p* = 0.416, η^2^ = 0.53, OP = 0.06; Td of the TA, *p* = 0.419, η^2^ = 0.10, OP = 0.20; Tc of the TA, *p* = 0.701, η^2^ = 0.02, OP = 0.31; Dm of the TA, *p* = 0.469, η^2^ = 0.04, OP = 0.08. Tukey post-hoc analysis for Td of the PL: CAI side vs. non-CAI side, *p* = 0.132, d = 0.85; CAI side vs. healthy, *p* = 0.037, d = 1.12; non-CAI side vs. healthy, *p* = 0.355, d = 0.59. Tukey post-hoc analysis for Tc of the PL: CAI side vs. non-CAI side, *p* = 0.122, d = 0.75; CAI side vs. healthy, *p* = 0.024, d = 1.04; non-CAI side vs. healthy, *p* = 0.712, d = 0.46.

**Table 1 healthcare-09-00707-t001:** Participant Demographic Information.

Variables	CAI Group (*n* = 12)	Healthy Group (*n* = 11)	*p* Value
Age (years)	21.4 ± 1.4	22.4 ± 1.4	n.s.
Sex (men:women)	11:1	9:2	n.s.
Height (cm)	172.4 ± 7.8	168.6 ± 8.3	n.s.
Mass (kg)	64.2 ± 8.7	60.8 ± 11.0	n.s.
Body mass index (kg/cm^2^)	21.6 ± 2.4	21.2 ± 2.4	n.s.
Cumberland Ankle Instability Tool	18.5 ± 4.7	28.1 ± 1.4	<0.001
Number of lateral ankle sprains	5.8 ± 3.2	0.0 ± 0.0	<0.001

Mean ± SD; CAI: chronic ankle instability; n.s.: non-significant.

**Table 2 healthcare-09-00707-t002:** Comparison of evertor and invertor mean normalized peak torques and the mean normalized maximum work done at 60°/s and 180°/s between the CAI and healthy groups of subjects.

Motion	Angular Velocity (°/s)	CAI Side (*n* = 12)	Non-CAI Side (*n* = 12)	Healthy (*n* = 11)	*p* Value	Post-Hoc	Effect Size	Observed Power
NPT (Nm/kg)								
Eversion	60	40.85 ± 14.18	42.32 ± 16.80	37.25 ± 8.23	n.s.	NA	0.03	1.00
	180	29.48 ± 9.23	27.39 ± 7.38	26.16 ± 4.83	n.s.	NA	0.01	0.94
Inversion	60	45.93 ± 14.15	48.13 ± 12.62	49.89 ± 16.25	n.s.	NA	0.04	0.99
	180	37.18 ± 8.80	34.91 ± 7.43	38.70 ± 12.71	n.s.	NA	0.03	0.98
NMW (J/kg)								
Eversion	60	35.46 ± 11.07	35.39 ± 11.32	34.41 ± 8.50	n.s.	NA	0.00	0.20
	180	24.70 ± 7.65	26.31 ± 7.16	27.31 ± 7.62	n.s.	NA	0.02	0.78
Inversion	60	39.60 ± 12.63	41.18 ± 11.55	42.72 ± 15.84	n.s.	NA	0.01	0.90
	180	32.34 ± 8.52	29.17 ± 7.63	34.87 ± 11.62	n.s.	NA	0.06	1.00

Mean ± SD; CAI: chronic ankle instability; NPT: normalized peak torque; NMW: normalized maximum work done; n.s.: non-significant; NA: not applicable.

**Table 3 healthcare-09-00707-t003:** Comparison of the muscle thickness between the CAI and healthy groups of subjects.

Muscle	CAI Side (*n* = 12)	Non-CAI Side (*n* = 12)	Healthy (*n* = 11)	*p* Value	Post-Hoc	Effect Size	Observed Power
PL thickness	2.12 ± 0.33	2.17 ± 0.27	1.94 ± 0.30	n.s.	NA	0.10	0.06
TA thickness	2.71 ± 0.42	2.83 ± 0.56	2.50 ± 0.29	n.s.	NA	0.09	0.06

Mean ± SD; CAI: chronic ankle instability; PL: peroneus longus; TA: tibialis anterior; n.s.: non-significant; NA: not applicable.

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
