# Peer review of "Tensiomyographic Neuromuscular Response of the Peroneus Longus and Tibialis Anterior with Chronic Ankle Instability"

_healthcare, 2021, doi:10.3390/healthcare9060707_

Round 1

Reviewer 1 Report

Thanks for the opportunity to review this interesting manuscript. Please, considere the following issues and commednations in order to improve the present manuscript:

1) For introduction, consider to ecpand your discussion about the distal and proximal muscle activation under chronick ankle instability.

Hum Mov Sci. 2017 Oct;55:211-220. 

2) In addition, the introduction should add information about US measurements under ankle sprains. Please, see:

 Nov-Dec 2016;39(9):635-644

3) The study design should be accurately detailed including study type and the followed guidelines according to STROBE criteria (please, see https://www.equator-network.org/)

Ann Intern Med. 2007; 147(8):573-577. PMID: 17938396

4) The methods should add the sample size calculation. This is a mjor issue that should be addressed.

5) Statistical analyses: Was post-hoc analysis carried out according to the Bonferroni`s correction?

6) Please, in addition to the Eta2 coeffficient, effect size should be completed for post-hoc comparisons by the Cohen d statistic.

7) Results: post.hoc analyses should be accurately detailed including P values of all comprarisons.

8) Discussion: Please, discuss if you measured the skin and subcutaneous tissue thickness in the TMG measurement points. This could influence your measurements:

2018 Jun;64(6):549-553.

Author Response

Dear Reviewer 1,

We would like to thank you for the critical appraisal and helpful suggestions you made on our manuscript. We believe that the quality of this manuscript has been improved by responding to the comments we received. Our responses to your comments are summarized point by point as attached. Changes made to the manuscript are highlighted. We believe that we have adequately addressed all the comments raised. 
Thank you very much.

Yours faithfully,
Tsubasa Tashiro

Graduate School of Biomedical and Health Sciences, Hiroshima University, Hiroshima, Japan
Post cord: 734-8553 
Address: 1-2-3 Kasumi, Minami-ku, Hiroshima, Japan
E-mail: tsubasatashiro716@hiroshima-u.ac.jp

Reviewer 2 Report

This study investigated muscle contractile response of the peroneus longus (PL) and tibialis anterior (TA) in groups with and without chronic ankle instability (CAI) using tensiomyography(TMG). They demonstrated the impaired muscle response and muscle contraction of PL in CAI patients, which represents the requirement of ankle evertor training for CAI.

1. How was the sample size calculated?

2. The severity of CAIT score may affect the parameters (i.e. TMG, muscle strength, and muscle thicknesses) and patient’s prognosis. I wonder if there was any difference in the parameter according to the severity of the CAIT score.

3. Are there any differences in the parameters according to the number of lateral ankle sprains?

4. In general, CAI is classified as mechanical and functional instability. Does your subject include both?

5. Classification of CAI may affect the parameters (i.e. TMG, muscle strength, and muscle thicknesses). I wonder if there was any difference in the parameters according to the classification?

6. Of some variables (the classification of CAI, number of lateral ankle sprains, CAIT score, et al), which one is most closely related to the TMG values of PL on the CAI side?

7. Can you perform a multivariate analysis for adjusting the demographic variables and give similar results?

8. Present the results from multiple tables as a single figure so that the reader can easily grasp the results of the study.

9. How many investigators measured the TMG parameters? 2 or 3? Please indicate the initials of the authors who measured the TMG parameters in the "Tensiomyography Assessment". Also, the intraclass correlation coefficient (ICC) for the TMG parameters needs to be calculated to investigate the interrater reliability between investigators.

10. I can't find Figure 1 in the manuscript.

11. The researcher who implemented Image J software to analyze ultrasound images needs to be blinded from the subject's information. Thus, describe whether it was blind or not in the "Ultrasound Measurement". Also, please indicate the initials of the authors who performed ultrasound measurements and image analysis.

12. On the CAI side, the TMG values ​​of PL were significantly higher in Td and Tc compared to healthy subjects. However, there was no statistically significant difference compared to non-CAI side. The reason needs to be described in the Discussion.

Author Response

Dear Reviewer 2,

We would like to thank you for the critical appraisal and helpful suggestions you made on our manuscript. We believe that the quality of this manuscript has been improved by responding to the comments we received. Our responses to your comments are summarized point by point as attached. Changes made to the manuscript are highlighted. We believe that we have adequately addressed all the comments raised. 
Thank you very much.

Yours faithfully,
Tsubasa Tashiro

Graduate School of Biomedical and Health Sciences, Hiroshima University, Hiroshima, Japan
Post cord: 734-8553 
Address: 1-2-3 Kasumi, Minami-ku, Hiroshima, Japan
E-mail: tsubasatashiro716@hiroshima-u.ac.jp

Round 2

Reviewer 1 Report

Authors have addressed my suggestions. Thanks.

Reviewer 2 Report

Thanks for the author's reply.